# The Distribution Pattern of First-Line Anti-Tuberculosis Drug Concentrations between the Blood and the Vertebral Focus of Spinal Tuberculosis Patients

**DOI:** 10.3390/jcm11185409

**Published:** 2022-09-15

**Authors:** Guanyin Jiang, Wanyuan Qin, Xing Du, Ye Zhang, Muzi Zhang, Tuotuo Xiong, Dezhang Zhao, Yunsheng Ou

**Affiliations:** 1Department of Orthopedics, The First Affiliated Hospital of Chongqing Medical University, Chongqing 400016, China; 2Orthopedic Laboratory, Chongqing Medical University, Chongqing 400016, China; 3College of Pharmacy, Chongqing Medical University, Chongqing 400016, China

**Keywords:** spinal tuberculosis, anti-tuberculosis treatment, drug concentration, distribution pattern, sclerotic bone

## Abstract

Background: Anti-tuberculosis drug concentrations are critical for the treatment of spinal tuberculosis. The distribution pattern of anti-tuberculosis drugs between the blood and the vertebral focus needs to be further explored. Methods: A total of 31 spinal tuberculosis patients were prospectively included and then divided into a sclerotic group (15 cases) and a non-sclerotic group (16 cases) according to their preoperative CTs. All patients were treated with 2HERZ/6H_2_R_2_Z_2_ chemotherapy for 4 weeks before the operation. During the operation, blood, normal vertebral bone tissue, and vertebral focus tissue were obtained, processed, and sent to the pharmacology laboratory. The concentration values of four anti-tuberculosis drugs in each sample were obtained in a pharmacology laboratory. Results: There was no significant difference in the concentrations of the four anti-tuberculosis drugs in the blood and the normal vertebral bone tissue between the two groups; however, there was a significant difference in the vertebral focus tissue. There existed a linear correlation of four anti-tuberculosis drug concentrations between the blood and the focus in the non-sclerotic bone group. Conclusions: The existence of sclerotic bone hinders the anti-tuberculosis drug distribution. In the absence of sclerotic bone in the vertebral focus, there exists a linear relationship of the four anti-tuberculosis drug concentrations between the blood and the vertebral focus of spinal tuberculosis patients.

## 1. Introduction

Tuberculosis is an infectious disease with a long history, and spinal tuberculosis (STB) has existed throughout human history. Fossils indicating cervical tuberculosis in the Neolithic age, from approximately 10,000 BC, have been unearthed in modern Germany [1]. Spinal tuberculosis is a special type of osteoarticular tuberculosis with high morbidity, accounting for approximately 50% of osteoarticular tuberculosis cases [2]. At present, anti-tuberculosis drug therapy combined with surgical treatment is considered the gold standard for STB patients with spinal structural damage, nerve compression symptoms, or an insensitivity to anti-tuberculosis chemotherapy [3]. The debridement of lesions is a key step in STB surgery as it enhances control over the tubercular changes, improves the efficacy of anti-tuberculosis drugs, promotes bone-graft fusion, and reduces the risk of the recurrence of STB [4]. Moreover, anti-tuberculosis drug therapy is critical for STB treatment, and it is essential, regardless of whether the patient receives surgical treatment [3].

Pharmacological studies have shown that the clinical efficacy of a drug depends on its concentration in the target tissue [5]. Several pharmacokinetic factors can affect the distribution of anti-tuberculosis drugs. Food can hinder the absorption of INH and RFP, while not that of PZA or EMB; therefore, it is recommended that patients take anti-tuberculosis drugs on an empty stomach [6]. The metabolic rate of INH is associated with patients’ genetic types, which ranges from 1.5 to 4 h, and the metabolic types of INH include the fast type (49.3%), the middle type (25.1%), and the slow type (25.6%) in China [7]. Interactions between other drugs and anti-tuberculosis drugs also matter. For example, aluminum hydroxide reduces the gastrointestinal absorption of EMB, and ammonia salicylic acid can affect the absorption of RFP [8]. Whether or not the concentration of anti-tuberculosis drugs in a patient’s spinal focus can achieve a minimum inhibitory concentration (MIC) is a problem about which clinicians are very concerned. However, there are few studies on the distribution of anti-tuberculosis drug concentrations in the vertebral focus of STB, which makes it difficult to evaluate the concentrations in the spinal tuberculosis focus.

According to pharmacokinetic studies, after being absorbed into the blood through the digestive tract, anti-tuberculosis drugs are gradually distributed in all of the organs and tissues of the body, along with circulating in the blood throughout the body [9].

Wen et al. found that there was a good linear relationship between the concentration of the second-line anti-tuberculosis drug linezolid in the blood and the concentration of the drug in the focus in patients with osteoarticular tuberculosis [10]. It can be speculated that there may also be a linear relationship in STB between the concentration of the first-line anti-tuberculosis drugs in the blood and that in the vertebral focus. Ge and Liu et al. detected the concentration of anti-tuberculosis drugs in the blood and in the vertebral focus of STB and found that the concentration in the focus showed a significant downwards trend compared with that in the blood [11,12]. In addition, when there was sclerotic bone in the vertebral focus, the downwards trend was more obvious, and the concentration of anti-tuberculosis drugs in the central area of the lesion could not even reach the MIC, which suggested that the presence of sclerotic bone may hinder the distribution of anti-tuberculosis drugs in the vertebra [12].

This study prospectively included suitable cases of STB patients who underwent debridement surgery in our hospital to determine the distribution pattern of anti-tuberculosis drugs between the blood and the vertebral focus.

## 2. Materials and Methods

This study was conducted in accordance with the Declaration of Helsinki (as revised in 2013) and was approved by the Institutional Ethics Board of The First Affiliated Hospital of Chongqing Medical University. All necessary written informed consent forms were provided by participants in this study.

### 2.1. Patient Selection

A total of 31 STB patients who underwent lesion debridement with a posterior approach fixation and a spinal fusion with a bone graft in our department, from January 2020 to September 2021, were prospectively included. The anti-tuberculosis drugs that the included patients took were isoniazid (INH), rifampin (RPF), pyrazinamide (PZA), and ethambutol (EMB). All patients were treated with 2HERZ/6H_2_R_2_Z_2_ chemotherapy for 4 weeks before the operation. The definition of 2HERZ/6H_2_R_2_Z_2_ is that, in the first 2 months, the STB patients take INH, RPF, PZA, and EMB once a day, and in the following 6 months, patients take INH, RPF, and PZA twice a week. All patients had fully recovered with no STB recurrence at the 1-year follow-up.

#### 2.1.1. Inclusion Criteria

Patients were selected if they met the following inclusion criteria: (1) vertebral focus tissues were extracted in surgery and confirmed as STB by pathological diagnosis; (2) they received preoperative, standardized anti-tuberculosis chemotherapy for at least 4 weeks; (3) they were initially treated with surgery; and (4) their complete medical records were available, including general information, perioperative laboratory examination results, and imaging results (including those from magnetic resonance imaging (MRI) and computed tomography (CT)).

#### 2.1.2. Exclusion Criteria

Patients were excluded if they presented with the following: (1) preliminary and pathological diagnoses of diseases other than STB; (2) a lack of standardized anti-tuberculosis chemotherapy before surgery; (3) with the presence of severe infection and tumor chemotherapy; or (4) a previous history of STB surgery.

### 2.2. Measures and Statistics

#### 2.2.1. Sample Collection and Processing

Blood, normal vertebral bone tissue, and vertebral focus tissue were collected simultaneously during surgery, typically 2 h after the oral administration of medication. The normal vertebral bone tissue and vertebral focus tissue were extracted by the same surgeon, and the anatomical diagram of the collection sites for the normal vertebral bone tissue and the vertebral focus tissue is shown in Figure 1. The venous blood samples were collected in glass tubes containing heparin anticoagulant by the nurse according to the given time phases. All of the samples were stored in a container of liquid nitrogen immediately upon extraction and sent to the laboratory for further processing.

The venous blood samples were centrifuged at 5000 rpm for 15 min to obtain the plasma for further determinations. The normal vertebral bone tissue and vertebral focus tissue were washed with saline and dried with a paper tissue before being weighed in the laboratory. The above-mentioned tissues from the vertebrae were cryodesiccated and pulverized into a fine powder using liquid-nitrogen-combined grinding. A powder of 200 μg was mixed with 2 mL of an extract of dimethylcarbinol-dichloromethane (1:1) and centrifuged for 15 min successively, and the supernatant was collected. The processing time of the plasma and supernatant samples was 30 min and 45 min, respectively. All of the samples were all stored at −80 °C.

#### 2.2.2. Apparatus

We used an Agilent Infinity 1290 high-performance liquid chromatography–tandem mass spectrometer (HPLC–MS/MS) (Agilent Technologies, Santa Clara, CA, USA); an XH-B vortex mixer (Jiangsu Healthcare Medical Supplies Co., Ltd., Danyang, China); an iCen 24 refrigerated centrifuge (Hangzhou Allsheng Instruments Co., Ltd., Hangzhou, China); a VORTEX 2 S025 constant-temperature culture oscillator (IKA, Staufen, Germany); and a BSA224S-CW electronic scale (Sartorius, Gottingen, Germany).

#### 2.2.3. HPLC–MS/MS Condition

For the chromatographic column, we used a Waters ACQUITY UPLC HSS T3 C18 (1.8 μm, 2.1 × 150 mm). The mobile phase A was a 2 mmol L^−1^ ammonium formate solution containing 0.1% formic acid; the mobile phase B was an acetonitrile solution containing 0.1% formic acid. The gradient elution program was conducted as follows: Mobile phase B was increased linearly from 0% to 100% over 7 min and kept at a ratio of 100% from 7 to 9.5 min; then, it was decreased linearly from 100% to 0% from 9.5 min to 9.6 min, and it was held at this level for 2 min so that the system could re-equilibrate before the next injection. The flow rate was 0.20 mL min^–1^; the column temperature was 35 °C; and the sample quantity was 1 μL. The mass spectrometry conditions were as follow: electrospray ionization multiply reaction monitoring mode (ESI-MRM); electrospray voltage: −4 kV; ion source temperature: 350 °C; gas temperature: 300 °C; and gas flow: 8 L min^−1^. The MRM transitions (m/z) of the four drugs were as follow: INH: 138.1→121.1; RIF: 823.4→791.4; PZA: 124.1→81.1; and EMB: 205.2→116.1.

#### 2.2.4. Statistical Analysis

Continuous variables are presented as the mean ± SD for normally distributed data. The relationships between the plasma and bone concentrations of the four anti-tuberculosis drugs was examined using Pearson’s correlation analysis. *P* < 0.05 was considered statistically significant. SPSS version 26.0 software (SPSS Inc., Chicago, IL, USA) was used for the statistical analyses.

## 3. Results

### 3.1. Patient Characteristics

Among the 31 STB patients enrolled in the study, 17 males and 14 females were divided into a sclerotic group (15 cases) and a non-sclerotic group (16 cases) (Table 1). The mean age of the patients in the sclerotic group was 45.87 ± 19.48, and that of the patients in the non-sclerotic group was 50.75 ± 17.09 years (Table 1). The 31 STB patients included 19 thoracic patients and 12 lumbar patients. There was no statistical significance in the perioperative characteristics between the two groups, including sex, age, body mass index (BMI), site of infectious focus, operation time, operation blood loss, course of disease, and anti-tuberculosis drug administration time (Table 1).

### 3.2. The Results of Anti-Tuberculosis Drug Concentrations in Each Group

A total of 4 anti-tuberculosis drug concentrations in 3 kinds of samples from the sclerotic bone group and the non-sclerotic bone group are listed in Table 2. We found the same drug concentration distribution trend in both groups: The 4 anti-tuberculosis drug concentration values in the blood were the highest, followed by those in the normal vertebral bone tissues, and the concentration values of the vertebral focus tissues were the lowest (Table 2). This anti-tuberculosis drug concentration distribution trend is consistent with that found by the previous studies reported by ZH Ge and P Liu et al. [12,13]

### 3.3. Comparison of Drug Concentrations in Different Samples between the Two Groups

There was no statistical significance found in any of the four anti-tuberculosis drug concentrations between the two groups in either the blood or the normal vertebral bone tissue (Figure 2a,b). However, in the vertebral focus tissue, all four anti-tuberculosis drug concentrations in the sclerotic bone group were lower than those in the non-sclerotic bone group, which was statistically significant (Figure 2c).

#### The Linear Relationship of Drug Concentrations in the Non-sclerotic Bone Group

In the absence of sclerotic bone in the vertebral focus, there was a linear relationship for all 4 anti-tuberculosis drug concentrations between the blood and the vertebral focus of the spinal tuberculosis patients (Figure 3). The R^2^ values for RPF, INH, PZA, and EMB were 0.955, 0.927, 0.813, and 0.809, respectively.

## 4. Discussion

Spinal tuberculosis is the most common type of osteoarticular tuberculosis, mostly secondary to tuberculosis [14]. Anti-tuberculosis chemotherapy is considered the basic treatment for STB, especially when STB patients require surgical treatment [15]. However, clinical studies have shown that non-standard anti-tuberculosis chemotherapy, including insufficient drug dose administration and insufficient chemotherapy course, is a key reason for postoperative STB recurrence [15,16]. Our surgeon found that, in clinical practice, some STB patients can obtain a clinical cure via the administration of standard anti-tuberculosis chemotherapy alone. Moreover, postoperative non-standard anti-tuberculosis chemotherapy can lead to STB recurrence even after surgery [17]. The above clinical phenomena suggest that it is of great significance to evaluate the efficacy of chemotherapy in STB patients to predict the prognosis, and the most intuitive way to evaluate that is to observe whether the concentration of anti-tuberculosis drugs in the focus reaches the MIC.

Measuring the anti-tuberculosis drug concentration in the vertebral focus is significant; however, it is difficult to perform in clinical practice because surgeons can obtain the vertebral focus tissue only via an operation, which is not applicable to STB patients who desire conservative treatment. Therefore, we have been working to find a new, secure method, via the drug blood concentration, to evaluate anti-tuberculosis drug concentrations in the STB vertebral focus in the absence of invasive surgery. Pharmacokinetics have proven that oral drugs must go through absorption and blood circulation and eventually reach target tissues or organs [18]. Therefore, there may be a linear correlation in the drug concentration between the blood and the target tissue, such as tuberculosis foci. It was reported by Wen et al. that the concentration of the second-line anti-tuberculosis drug linezolid in the bone had a linear correlation with the drug concentration in the plasma in osteoarticular tuberculosis patients [10]. Meanwhile, a previous study found that linezolid had good penetrability into diseased bone tissue in multiple-drug-resistant STB patients, which means that linezolid concentrations in the plasma were positively related to those in the diseased bone tissue [19]. These studies are supportive of our findings and further suggest that, due to the linear correlation, it is feasible and scientific to predict drug concentration in the STB focus via the immediate drug concentration in the blood.

In this study, we used HPLC–MS/MS to analyze the concentration distribution pattern of four anti-tuberculosis drugs between the blood and the vertebrae in STB patients. Compared with previous studies, this study is unique in measuring the distribution of multiple anti-tuberculosis drugs in different tissues within the same sample [12,13,20]. The results show that, in the sclerotic bone group, the average concentrations of INH, RPF, PZA, and EMB in vertebral focus tissue were 0.01 ± 0.011 μg/mg, 0.094 ± 0.015 μg/mg, 0.245 ± 0.364 μg/mg, and 0.086 ± 0.074 μg/mg, respectively. The MIC of INH, RFP, and EMB have thus far been known as 0.025–0.05, 0.005–0.5, and 0.5–8 ug/mL, respectively, and the MIC of PZA has been measured at 1.5 ug/mL under the condition of a pH of 5.0 [8]. It is widely accepted that an effective bactericidal concentration level should be 10 times that of the MIC level [21]. The average concentrations of INH, RPF, PZA, and EMB in vertebral focus tissue were all lower than the corresponding effective bactericidal concentration, and most were lower than the corresponding MIC, which suggests that the effect of conservative treatment may be not good for those patients with sclerotic bone in the vertebral focus. Therefore, surgical debridement should be performed on patients with sclerotic bone in the vertebral focus [12,13]. However, due to bone-tissue-sample processing limitations, the drug concentration in vertebral focus tissue may be an underestimate, an issue which needs to be further validated in future work.

In the absence of sclerotic bone in the vertebral focus, there was a linear correlation in the concentration of the four anti-tuberculosis drugs between the blood and the vertebral focus, which means that the higher the drug concentration in the blood was, the higher the drug concentration in the vertebral focus was. The R-square value of the linear correlation of the 4 drugs was more than 80%, which suggests that using blood drug concentration to predict the vertebral focus drug concentration is possible in STB. This finding has great clinical significance, especially in assisting with the clinicians’ assessment of STB patients. First, it is widely accepted that, for STB patients with no symptoms of nerve compression or obvious spinal mechanical structure damage, conservative anti-tuberculosis drug therapy is recommended to achieve full recovery [22,23]. Unfortunately, some of the aforementioned SBT patients cannot achieve satisfactory recovery due to several factors, including non-standard chemotherapy plans, poor compliance, insufficient drug digestive absorption, and others [24,25,26]. For this kind of patient, particularly those without sclerotic bone in the vertebral focus, we can use the blood drug concentration to predict the vertebral focus drug concentration by the linear correlation found in our study and evaluate the effect of the anti-tuberculosis chemotherapy instantly to adjust the administration plan and even the therapeutic strategy. The suitable use of the linear correlation can improve the safety of anti-tuberculosis drug therapy and affect the deterioration of some STB patients over time. In addition, for STB patients who have undergone surgical debridement, this study may be helpful in monitoring the efficacy of anti-tuberculosis drugs after surgery. There is no sclerotic bone in the vertebral focus after surgical debridement; therefore, the postoperative concentration of anti-tuberculosis drugs in the vertebral focus can also be estimated by the blood drug concentration to evaluate the efficacy of anti-tuberculosis drugs after surgery. This measure can enrich the evaluation methods of the postoperative anti-tuberculosis effect and help to reduce recurrence and drug-resistant tuberculosis occurrence. However, because of the complete debridement of the vertebral focus, the surgery may lead to changes in the blood transport system in the affected vertebra anatomically, thus affecting the distribution of anti-tuberculosis drugs. The linear relationship of anti-tuberculosis drug concentration needs to be further verified. If the distribution pattern can be further explored, it will be of great significance to the rehabilitation of STB patients and the prevention of tuberculosis recurrence

## 5. Conclusions

The existence of sclerotic bone hinders anti-tuberculosis drug distribution. In the absence of sclerotic bone in the vertebral focus, there exists a linear relationship of the four anti-tuberculosis drug concentrations between the blood and the vertebral focus of spinal tuberculosis patients.

## Figures and Tables

**Figure 1 jcm-11-05409-f001:**
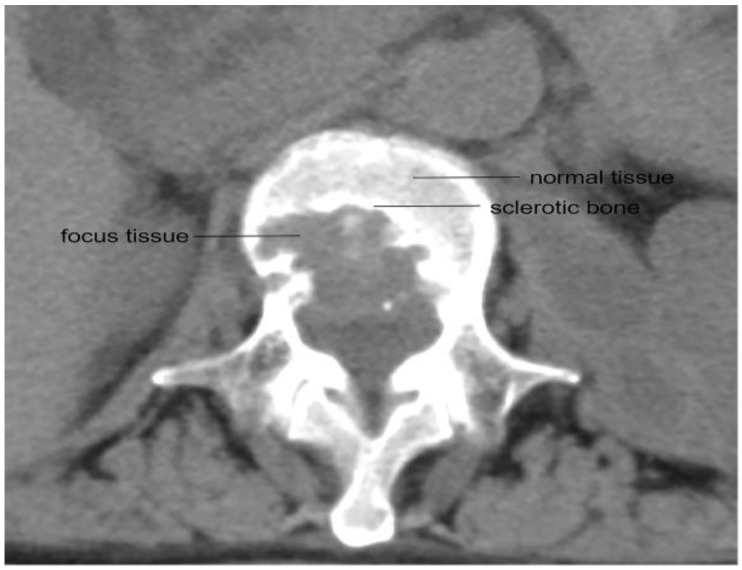
CT scan showing the location of healthy bone tissue and the focus tissue in a vertebra.

**Figure 2 jcm-11-05409-f002:**
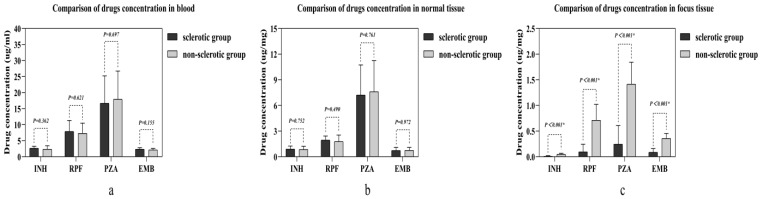
Comparison of four drug concentrations in different samples between the two groups: (**a**) the concentration comparison in blood; (**b**) the concentration comparison in normal tissue; (**c**) the concentration comparison in focus tissue. Abbreviations: isoniazid (INH), rifampin (RPF), pyrazinamide (PZA), and ethambutol (EMB). The independent samples *t*-test was used in making the comparison. “*” means statistic significance.

**Figure 3 jcm-11-05409-f003:**
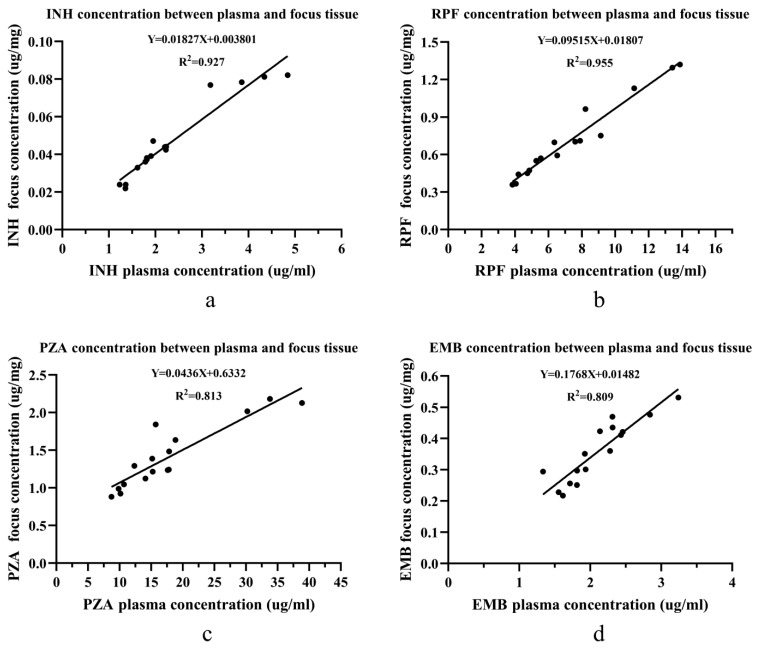
The linear relationship of four drug concentrations between the plasma and the focus tissue in the non-sclerotic bone group: (**a**) the linear relationship for INH; (**b**) the linear relationship for RPF; (**c**) the linear relationship for PZA; and (**d**) the linear relationship for EMB. Abbreviations: isoniazid (INH), rifampin (RPF), pyrazinamide (PZA), and ethambutol (EMB). Linear regression was used to find the concentration relationship between the plasma and the focus tissue.

**Table 1 jcm-11-05409-t001:** Perioperative characteristics of 31 thoracic and lumbar tuberculosis patients.

Characteristics	Sclerotic Group (n = 15)	Non-Sclerotic Group (n = 16)	*p* Value
Age (year)	45.87 ± 19.48	50.75 ± 17.09	0.463
BMI (kg/m^2^)	21.37 ± 2.81	20.70 ± 3.09	0.630
Sex (n, %)			0.870
Female	7 (46.7%)	7 (43.8%)	
Male	8 (53.3%)	9 (56.2%)	
Site of infectious focus (n)			0.886
Thoracic	9	10	
Lumbar	6	6	
Operation time (min)	191.20 ± 29.98	185.19 ± 27.64	0.566
Operation blood loss (mL)	343.93 ± 174.62	330.50 ± 208.43	0.859
Course of disease (month)	5.57 ± 4.15	6.13 ± 3.14	0.730
Administration time (day)	25.80 ± 3.97	27.19 ± 4.22	0.354

Abbreviation: Body mass index (BMI).

**Table 2 jcm-11-05409-t002:** The comparison of anti-tuberculosis drug concentrations in plasma, healthy bone tissues, and the focus between the sclerotic bone group and the non-sclerotic bone group.

Anti-Tuberculosis Drugs	Plasma (μg/mL)			Healthy Bone Tissue (μg/mg)			Focus Tissue (μg/mg)		
	Non-S Group	S Group	*p*-Value	Non-S group	S Group	*p*-Value	Non-S Group	S Group	*p*-Value
Isoniazid	2.356 ± 1.101	2.656 ± 0.609	0.362	0.849 ± 0.371	0.891 ± 0.361	0.752	0.047 ± 0.021	0.010 ± 0.011	<0.001 *
Rifampin	7.287 ± 3.197	7.884 ± 3.449	0.621	1.776 ± 0.772	1.940 ± 0.487	0.490	0.711 ± 0.311	0.094 ± 0.150	<0.001 *
Pyrazinamide	17.928 ± 8.838	16.699 ± 8.536	0.697	7.611 ± 3.638	7.215 ± 3.531	0.761	1.415 ± 0.427	0.245 ± 0.364	<0.001 *
Ethambutol	2.107 ± 0.496	2.358 ± 0.457	0.155	0.738 ± 0.352	0.734 ± 0.361	0.972	0.358 ± 0.098	0.086 ± 0.074	<0.001 *

Abbreviations: Non-sclerotic group (Non-S group); sclerotic group (S group). * indicates statistical significance.

## Data Availability

The data presented in this study are available on request from the corresponding author.

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
