# Peer review of "The Distribution Pattern of First-Line Anti-Tuberculosis Drug Concentrations between the Blood and the Vertebral Focus of Spinal Tuberculosis Patients"

_jcm, 2022, doi:10.3390/jcm11185409_

Round 1

Reviewer 1 Report

Jiang et al quantitate the concentration of first-line anti-TB drugs in plasma and vertebral focus tissue. The authors make two observations: 1) Drug concentrations at the focus tissue in sclerotic bone is lower than in non-sclerotic bone (and in both cases, these values are lower than that found in plasma); and 2) There is a linear correlation in non-sclerotic group between plasma and focus concentration (though data on the relationship between plasma and focus concentration for the sclerotic group is never given). The manuscript however suffers from a number of errors: 1) Concentration of drugs is presented as μl/mg which makes no sense – drug concentration is usually expressed as a unit of mass per volume; 2) Figure 2, y-axis is not labeled; 3) Table 1, ‘Sex(n,%)’ row only has the n-value not the %; 4) Figure 3, units are not given for y- and x-axes.

Additionally, no data on treatment response/clinical outcomes of the patients is given. For example, authors state the following but provide no data to support it: “The results showed that in the sclerotic bone group, the concentrations of the four anti-tuberculosis drugs in vertebral focus tissue were all lower than the corresponding MBC. The effect of conservative treatment was not good for these patients with sclerotic bone in the vertebral focus; moreover, they were prone to drug-resistant tuberculosis.” Also statistics are missing from Tables 2-3, which I recommend combining into one table and highlighting statistically significant differences.

As for tables 2-3, I do not quite understand how normalization is performed between plasma (a liquid) and tissue (a solid) for purposes of determining drug concentration. Further, for blood the sample is spun down and only plasma used, which concentrates the amount of drug since red/white blood cells are removed; however for tissue such as bone (where the majority of mass comes from hydroxyapatite) no such depletion is performed and thus an underestimation would be expected since I presume it’s normalized to weight (again the mislabeling of the units in this case prevents any firm conclusions one way or another). I suppose decalcification/collagenase treatment might help solve this which brings me to my next point: Another variable is the half-life of each of the drugs and how the different methods used to prepare the tissue types for HPLC‒MS/MS (ie plasma vs bone) might affect drug degradation. Please in your Materials and Methods comment on this, and include pertinent info such as the total amount of time required for storage and preparation of the different sample types for HPLC‒MS, and at what temperature each step was performed at.

It would be beneficial to state the MBCs and MICs for the four first-line drugs somewhere in the manuscript so the reader has a frame of reference. For example, even though the concentrations were higher in non-sclerotic tissue, were these higher than the MBCs/MICs for each drug?

Author Response

1.Concentration of drugs is presented as μl/mg which makes no sense – drug concentration is usually expressed as a unit of mass per volume

Responds: Thanks for your comment. We are sorry about the writing mistake of the concentration units. The plasma concentration unit should be in µg/ml and bone tissue concentrations unit should be in µg/mg, we have corrected the units in the manuscript.

2.Figure 2, y-axis is not labeled. Table 1, ‘Sex(n,%)’ row only has the n-value not the %. Figure 3, units are not given for y- and x-axes

Responds: Thanks for your comments. We have labeled y-axis in the Figure 2, added the percent value of Sex in Table 1, and attached the concentration units for the y- and x-axes in Figure 3.

3.No data on treatment response/clinical outcomes of the patients is given. For example, authors state the following but provide no data to support it: “The results showed that in the sclerotic bone group, the concentrations of the four anti-tuberculosis drugs in vertebral focus tissue were all lower than the corresponding MBC.The effect of conservative treatment was not good for these patients with sclerotic bone in the vertebral focus; moreover, they were prone to drug-resistant tuberculosis.” Also statistics are missing from Tables 2-3, which I recommend combining into one table and highlighting statistically significant differences.

Responds: Thanks for your comments. We have added the data in our revised manuscript to support the clinical outcomes of the included STB patients like “The results showed that in the sclerotic bone group, the average concentrations of INH, RPF, PZA and EMB in vertebral focus tissue are 0.01 ± 0.011 μg/mg, 0.094 ± 0.015 μg/mg, 0.245 ± 0.364 μg/mg and 0.086 ± 0.074 μg/mg separately. The MIC of INH, RFP and EMB so far have been known as 0.025-0.05, 0.005-0.5 and 0.5-8 ug/ml, respectively, and the MIC of PZA related cellular power of hydrogen is 1.5 ug/ml in the condition of pH 5.0 (1). The average concentrations of INH, RPF, PZA and EMB in vertebral focus tissue were all lower than the corresponding MBC, which means the effect of conservative treatment was not good for these patients with sclerotic bone in the vertebral focus; moreover, they were prone to drug-resistant tuberculosis.” Besides that we have combined the Table 2 and Table 3 as one table (the new Table 2) and added the static results in it.

Reference

[1] Lee J (1997) Clinical pharmacology, 2 edn. People’s Health Publishing House, Beijing, pp 692–700

4.As for tables 2-3, I do not quite understand how normalization is performed between plasma (a liquid) and tissue (a solid) for purposes of determining drug concentration.

Responds: Thanks for your comments. We are sorry about the confusion caused to you if the “normalization is performed between plasma (a liquid) and tissue (a solid)” means that the concentration units of plasma and bone tissue is the same in our original manuscript. We are sorry about the writing mistake and the insufficient quality checking work of our manuscript, which is going to be greatly improved in our revised manuscript. The plasma concentration unit should be in µg/ml and bone tissue concentrations unit should be in µg/mg, we have corrected the units in the manuscript. The result of the normalization between plasma and tissue is: 1 µg/ml = 1 µg/mg, which is measured by conversion. In our sample processing procedure, we mixed 200 µg bone tissue powder with 2 ml of the extract of dimethylcarbinol-dichloromethane to get the bone homogenate (dilution ratio is 1: 10), the bone tissue concentration of the bone homogenate is 100 µg/ml, which means 1ml bone homogenate containing 100 µg bone tissue. The unit of drugs concentration by HPLC/MSMS in the bone tissue is µg/mg, which is also 0.001 µg/µg that means 1 µg bone tissue containing 0.001 µg drug. Therefore 1ml bone homogenate containing 100ug bone tissue which containing 0.1 µg drug, the drug concentration of bone homogenate is 0.1 µg/ml. Finally considering that the dilution ratio of bone tissue is 1: 10, we can concluded that 1 µg/ml = 1 µg/mg.

5.Further, for blood the sample is spun down and only plasma used, which concentrates the amount of drug since red/white blood cells are removed; however for tissue such as bone (where the majority of mass comes from hydroxyapatite) no such depletion is performed and thus an underestimation would be expected since I presume it’s normalized to weight (again the mislabeling of the units in this case prevents any firm conclusions one way or another). I suppose decalcification/collagenase treatment might help solve this.

Responds: Thanks for your comments. We are totally agree with your opinion that the difference in depletion may cause the concentration underestimation in the bone tissue especially when its not normalized to weight (we are sorry about the errors in the units which is now corrected in the manuscript). To reduce the difference in depletion step of bone tissue, we once extracted the the supernatant of the bone power with dimethylcarbinol-dichloromethane (1:1) in this study. Thanks to your valuable advice, decalcification/collagenase treatment is a promising chose for our future study to improve our measurement quality.

6.which brings me to my next point: Another variable is the half-life of each of the drugs and how the different methods used to prepare the tissue types for HPLC‒MS/MS (ie plasma vs bone) might affect drug degradation. Please in your Materials and Methods comment on this, and include pertinent info such as the total amount of time required for storage and preparation of the different sample types for HPLC‒MS, and at what temperature each step was performed at.

Responds: Thanks for your comments. The half-life of these anti-tuberculosis drugs have great influence to the measurement results, the time of half-life of INH, RPF, PZA and EMB is 3.1 ± 1.1, 3.5 ± 0.8, 9~10 and 9~10 hours, respectively. The time that the blood drug concentration reached the peal value after administration of four anti-tuberculosis drugs is 1-2, 2-4, 2-4 and 2 hours for INH, RPF, PZA and EMB, respectively. Considering the above two aspects, we chose to extract the blood and bone tissue samples simultaneously during the debridement surgery in 2 hours after the preoperative drugs administration and then send the samples to be stored at -80  immediately, which can make the blood drug concentration reaches the peak value and no longer than the half-life time so as to decrease the effect of decomposition. As for the different process method for preparing plasma sample and bone tissue sample, we are sorry that we didnt make it clearly in the manuscript. We have added the specific process procedures in the revised manuscript containing the total time for storage and preparation of the different sample types for HPLC‒MS, and the corresponding temperature at each step.

7.It would be beneficial to state the MBCs and MICs for the four first-line drugs somewhere in the manuscript so the reader has a frame of reference. For example, even though the concentrations were higher in non-sclerotic tissue, were these higher than the MBCs/MICs for each drug?

Responds: Thanks for your comments. We are agree with your opinion that MBCs and MICs for the four first-line drugs are beneficial to build the frame of reference to our study conclusion. The MIC of INH, RFP and EMB so far have been known as 0.025-0.05, 0.005-0.5 and 0.5-8 ug/ml, respectively, and the MIC of PZA related cellular power of hydrogen is 1.5 ug/ml in the condition of pH 5.0 (1). Our results suggest that average concentrations of INH, RPF, PZA and EMB in vertebral focus tissue of non-sclerotic group were all higher than the corresponding MBC.

Reference

[1] Lee J (1997) Clinical pharmacology, 2 edn. People’s Health Publishing House, Beijing, pp 692–700

Reviewer 2 Report

Title: “The” before first line anti Tb drugs should be removed so the title reads “The distribution pattern of first-line anti-tuberculosis drugs concentration between blood and vertebral focus of spinal tuberculosis patients”

Abstract

Line 24:  Should read “between the two groups, however, there was a significant difference in the vertebral focus tissue.

Introduction

Line 49: Dr Wen et al, should be Wen et al

Comment 1: The authors set out to determine the distribution pattern of anti-tuberculosis drugs between blood and vertebral focus, however in the introduction they mention that “Ge and Liu et al. detected the concentration of anti-tuberculosis drugs in the blood and in the vertebral focus of STB and found that the concentration in the focus showed a significant downwards trend compared with that in the blood [7, 8]. What they set out to do had already been done by Ge and Liu so can they explain what is unique about their study.

Comment 2: From line 57 to 65, the findings mentioned in Ge et al., 2008 are similar to what the present authors also reported from their study, so again the question is, if the fact that the presence of sclerotic bone in the vertebral focus affects the concentration of anti-tuberculosis drugs has already been established, what was the reason for the present study, was it merely to confirm the earlier findings by the previous authors?

Comment 3: The authors conclude that “the existence of sclerotic bone hinders the anti-tuberculosis drug distribution” this was not a new finding as the introduction makes it clear that this was already known. What is the practical implication or treatment options based on this finding with regards to treatment of STB in patients with sclerotic bone

Author Response

1.Title: “The” before first line anti Tb drugs should be removed so the title reads “The distribution pattern of first-line anti-tuberculosis drugs concentration between blood and vertebral focus of spinal tuberculosis patients”. Abstract: Line 24:  Should read “between the two groups, however, there was a significant difference in the vertebral focus tissue. Introduction: Line 49: Dr Wen et al, should be Wen et al

Responds: Thanks for your efforts to help improve the writing quality of this manuscript. We have revised the manuscript in the title, abstract and introduction as advised.

2.The authors set out to determine the distribution pattern of anti-tuberculosis drugs between blood and vertebral focus, however in the introduction they mention that “Ge and Liu et al. detected the concentration of anti-tuberculosis drugs in the blood and in the vertebral focus of STB and found that the concentration in the focus showed a significant downwards trend compared with that in the blood [7, 8]. What they set out to do had already been done by Ge and Liu so can they explain what is unique about their study.

Responds: Thanks for your comment. We are sorry about that we didnt clearly put forward the unique novelty of our study in the manuscript, which may cause confusion to the reviewers. There are the novelties of this study: (a) We have found that the drugs concentration hinder effect of the sclerotic bone is partial, rather than complete which Ge and Liu et al. found in previous study [1, 2], due to the more precise measurement method HPLC/MSMS. (b) In the absence of the sclerotic bone, we have found that there exists a linear relationship of anti-tuberculosis drug concentrations between the blood and the vertebral focus, which demonstrably reveals the anti-tuberculosis drug concentrations distribution pattern in STB patients without sclerotic bone. (c) In our study we included four anti-tuberculosis drugs, which is more than three kinds of anti-tuberculosis drugs in previous studies.

Reference

[1] Ge Z, Wang Z, Wei M. Measurement of the concentration of three antituberculosis drugs in the focus of spinal tuberculosis. Eur Spine J. 2008 Nov;17(11):1482-7. doi: 10.1007/s00586-008-0778-7. Epub 2008 Sep 16. PMID: 18795341; PMCID: PMC2583188.

[2] Liu P, Zhu Q, Jiang J. Distribution of three antituberculous drugs and their metabolites in different parts of pathological vertebrae with spinal tuberculosis. Spine (Phila Pa 1976). 2011 Sep 15;36(20):E1290-5. doi: 10.1097/BRS.0b013e31820beae3. PMID: 21311403.

3.From line 57 to 65, the findings mentioned in Ge et al., 2008 are similar to what the present authors also reported from their study, so again the question is, if the fact that the presence of sclerotic bone in the vertebral focus affects the concentration of anti-tuberculosis drugs has already been established, what was the reason for the present study, was it merely to confirm the earlier findings by the previous authors?

Responds: Thanks for your comment. The origin purpose of our series studies is to establish a prediction model using plasma drug concentration to predict the vertebral focus drug concentration in STB patients. Therefore we design this study to explore the drug concentration distribution pattern between blood and vertebral focus, and the sclerotic bone is the important effective factor which should be took into the consideration. Then in our study we did validate the conclusion that the presence of sclerotic bone in the vertebral focus affects the concentration of anti-tuberculosis drugs which Ge et al. reported in 2008. However we have found that the drugs concentration hinder effect of the sclerotic bone is partial, rather than complete which Ge and Liu et al. found in previous study, due to the more precise measurement method HPLC/MSMS we used rather than HPLC Ge and Liu et al. used. This is the new finding compared with previous studies and is also the next direction we will focus on.

4.The authors conclude that “the existence of sclerotic bone hinders the anti-tuberculosis drug distribution” this was not a new finding as the introduction makes it clear that this was already known. What is the practical implication or treatment options based on this finding with regards to treatment of STB in patients with sclerotic bone

Responds: Thanks for your comment. We are sorry about that we didnt clearly put forward the new finding about the hinder effect of the sclerotic bone in the manuscript. Ge and Liu et al. found the drugs concentration hinder effect of the sclerotic bone is complete because the results of drugs concentration in the vertebral focus are undetected by the HPLC. However thanks to the HPLC/MSMS, our study found that in the vertebral focus of sclerotic group, the anti-tuberculosis drug concentrations can be detected, which means the drugs concentration hinder effect of the sclerotic bone should be partial rather than complete. This is the new finding in our study and it suggests that in the presence of sclerotic bone, anti-tuberculosis drugs can partially penetrate into the vertebral focus and produce the limited cure efficacy. Based on this new finding, for STB patients with sclerotic bone the anti-tuberculosis chemotherapy alone wont be as effective as those without sclerotic bone, anti-tuberculosis chemotherapy combined with surgery is the more promising treatment strategy for this kind of STB patients, especially when they suffered spinal structural damage, nerve compression symptoms or insensitive to anti-tuberculosis chemotherapy. This the most important practical implication based on this finding.

Reviewer 3 Report

This is an important manuscript for TB clinicians working in this filed, but it lacks bioanalytical information to support these findings.  Plasma concentration units should be in µg/mL and bone concentrations in µg/g (then plasma concentrations converted for regression correlation).  A supplementary section describing the analytical method and a high-level validation summary should be included to support these findings.  Also, which international validation guidelines were followed?

Fig 2:  Include concentration with units on y-axis.

Table 2:  Update units according to comment above. 

Fig 3:  include concentration units.

Table 2:  update units.

Author Response

1.This is an important manuscript for TB clinicians working in this filed, but it lacks bioanalytical information to support these findings. 

Responds: Thanks for your approval to the importance of our manuscript.

2.Plasma concentration units should be in µg/mL and bone concentrations in µg/g (then plasma concentrations converted for regression correlation).

Responds: Thanks for your comment, we are sorry about the writing mistake and we have corrected the concentration units in the manuscript and then converted the plasma concentrations for regression correlation.

3.A supplementary section describing the analytical method and a high-level validation summary should be included to support these findings. Also, which international validation guidelines were followed?

Responds: Thanks for your comment, we have revised the manuscript to describe the analytical method more specific in each step and the validation of the method is also added in the manuscript. Our study is strictly followed the international guideline Good Clinical Practice (GCP) and the guidelines for validation of quantitative analysis methods of biological samples (1).

Reference

[1] Chinese Pharmacopoeia Commission (2020) Pharmacopoeia of The People’s Republic of China, Chinese Medical Science Press, Beijing, pp 466-471.

4.Fig 2: Include concentration with units on y-axis. Table 2: Update units according to comment above. Fig 3: include concentration units. Table 2: update units.

Responds: Thanks for your comment. We have included concentration with units on y-axis in Fig 2 and Fig 3, and updated the units in Table 2 and Table 3 in the manuscript as advised.

Reviewer 4 Report

- Abstract: to remove numbering in front of each section. What does 2HERZ/6H2R2Z2 means? Please define.

- Introduction: To elaborate more on the treatment of STB, particularly the first line of anti-TB. Also, to elaborate more on the pharmacokinetic factors that affect the distribution of anti-TB drugs into the target tissues including genetic predisposition of CYP, metabolism, drug-drug interaction, etc.

- Methods: how was the sample size calculated? For mass spectrometry, did you use positive or negative ionization mode? What types of test was used to compare the significance between groups? 

- Results: Table 1: sex should be presented as n and %, site of infection focus is it presented as n or %. Figure 2: should add the unit to "y axis" and define all the abbreviations in the legend as well as describe the data analysis and test used for significance. Figure 3: in the legend describe the statistical analysis used. Line 134: should be anti-tuberculosis drug administration time.

Author Response

1.Abstract: to remove numbering in front of each section. What does 2HERZ/6H2R2Z2 means? Please define.

Responds: Thanks for your comment. We have removed the numbering in front of each section in abstract. We are sorry about that in manuscript we didnt clearly clarify the anti-tuberculosis chemotherapy plan of the STB patients. The 2HERZ/6H2R2Z2 is the abbreviation of one standard anti-tuberculosis chemotherapy plan, which means in the first 2 months patients take isoniazid, rifampin, pyrazinamide and ethambutol once a day, and in the next 6 months patients take isoniazid, rifampin and pyrazinamide twice a week. The 2HERZ/6H2R2Z2 chemotherapy plan is recommend by Chinese Society of Tuberculosis of Chinese Medical Association [1]. We have add the definition in the manuscript.

Reference

[1] Diagnosis and Treatment Guide of Tuberculosis in China. Chin J Tuberc Respir 2: 5-9

2.Introduction: To elaborate more on the treatment of STB, particularly the first line of anti-TB. Also, to elaborate more on the pharmacokinetic factors that affect the distribution of anti-TB drugs into the target tissues including genetic predisposition of CYP, metabolism, drug-drug interaction, etc.

Responds: Thanks for your comment. We are agree with your point that anti-tuberculosis drugs treatment for STB and pharmacokinetic factors which affect the distribution of anti-tuberculosis drugs are both important to our studys background, which should be clearly elaborated in the introduction part. Anti-tuberculosis drug therapy combined with surgical treatment is considered the gold standard for STB patients with specific symptoms [1]. The debridement of lesions is a key step in STB surgery as it enhances the control of tuberculosis changes, improves the efficacy of anti-tuberculosis drugs, promotes bone graft fusion, and reduces the risk of recurrence of STB [2]. There are several pharmacokinetic factors that can affect the distribution of anti-tuberculosis drugs. Food can hinder the absorption of INH and RFP while not for PZA or EMB, therefore its recommended to undertake anti-tuberculosis drugs with an empty belly [3]. The metabolism rate of INH is associated with patients genetic type which among 1.5 to 4 hours and the metabolism type of INH including fast type (49.3%), middle type (25.1%) and slow type (25.6%) in China [4]. The interaction of other drugs and anti-tuberculosis drugs also maters. For example, aluminum hydroxide reduces gastrointestinal absorption of EMB and ammonia salicylic acid can affect the absorption of RFP [5]. We have added the corresponding elaborations in the introduction part.

Reference

[1] Wang YX, Zhang HQ, Li M, et al. Debridement, interbody graft using titanium mesh cages, posterior instrumentation and fusion in the surgical treatment of multilevel noncontiguous spinal tuberculosis in elderly patients via a posterior-only. Injury 2017 Feb;48(2):378-383.

[2] Boachie-Adjei O, Papadopoulos EC, Pellisé F, et al. Late treatment of tuberculosis-associated kyphosis: literature review and experience from a SRS-GOP site. Eur Spine J 2013 Jun;22 Suppl 4(Suppl 4):641-6.

[3] Zhao JJ, Lu Y. Research and progress of PK/PD for anti-tuberculosis drugs. Chin J Antituberc. Jun 2019; 41(06):700-704.

[4] Du BY, Yan BY, Xu MY, Chen DZ. Studies on the method of Microbioassay and Inactivation Type of Isoniazid in Serum. Chinese Journal of Tuberculosis and Respiratory Diseases. 1981;04(01):39-42.

[5] Lee J (1997) Clinical pharmacology, 2 edn. People’s Health Publishing House, Beijing, pp 692–700.

3.Methods: how was the sample size calculated? For mass spectrometry, did you use positive or negative ionization mode? What types of test was used to compare the significance between groups? 

Responds: Thanks for your comment. We had estimated the sample size before we conducted this study and confirmed that 24 patients could meet the statistical criteria.

We used the sample size estimation formula for clinical diagnosis study to estimate how many patients need to be included [1].

The formula was as follows:

In this formula, n represents the estimated sample size, rrepresents the correlation index, “α” represents the level of significance, Z1-α/2 is the bilateral Z value corresponding to the specified test level α, and Z1-β is the unilateral Z value corresponding to the specified class Ⅱ error. Before conducting our study, we set r=0.80, α=0.05 and β=0.2, thus we calculated n=12. Therefore, a total of 24 patients (12×2=24) need to be included.

In our study, for mass spectrometry we used the multi-reaction monitoring mode and the positive ionization mode. The statistical tests used to compare the significance between groups including: (a) Mann-Whitney rank sum tests or t-tests were used to compare the measurement data between the groups; (b) Chi-square test were used to compare categorical variables between the groups. P < 0.05 was considered statistically significant.

Reference

[1] Sokal RR, Rohlf FJ. 1995. Biometry. New York: W.H. Freeman and Company. 887

  1. Results: Table 1: sex should be presented as n and %, site of infection focus is it presented as n or %. Figure 2: should add the unit to "y axis" and define all the abbreviations in the legend as well as describe the data analysisand test used for significance. Figure 3: in the legend describe the statistical analysis used. Line 134: should be anti-tuberculosis drug administration time.

Responds: Thanks for your comment. We have revised the Table 1 as request and added the unit to "y axis" in Figure 2 with the definition of all the abbreviations and data analysis description in the legend. As for Figure 3, we have added the statistical analysis in the legend. In the line 134 we have revised the sentence as request.

Round 2

Reviewer 1 Report

This is a vastly improved version of the previous version. I commend the authors for making the recommended changes, which as improved the clarify of the manuscript. I only have a couple comments, the first being the most important as it relates to conclusions made in the manuscript which may not be supported by the data:

Major comment: In the below sentence you compares MICs of the drugs to concentrations of the same drugs found in sclerotic tissue. My concern is that MIC are presented in units of ug/ml whereas drug concentrations in sclerotic tissue are presented in different units (ug/mg). Since bone tissue consists of mostly dead weight in my opinion (calcium-phosphate) the effective concentration of the drug is likely much higher. Therefore I'm not sure of the soundness of making a direct comparison between known MICs and concentration of drug found in sclerotic tissue to conclude that the drugs are ineffective. Further you write that "The effect of conservative treatment was not good for these patients with sclerotic bone in the vertebral focus; moreover, they were prone to drug-resistant tuberculosis” -- but in the materials and method you write, "All patients have been fully 94 recovery with no STB recurrence after one year fellow-up.” These two sentences are incompatible. However, the fact that all patients fully recovered does support the notion that the effective concentration of anti-TB drugs in sclerotic tissue was sufficient for curative treatment, which goes against your hypothesis. Please explain. Finally, for the sentence ""they were prone to drug-resistant tuberculosis" where is the data in the paper to support this? 

Here is the paragraph I'm referring to:

"The results showed that in the sclerotic bone group, the average concentrations of INH, RPF, PZA and EMB in vertebral focus tissue are 0.01 ± 0.011 μg/mg, 0.094 ± 0.015 μg/mg, 0.245 ± 0.364 μg/mg and 0.086 ± 0.074 μg/mg separately. The MIC of INH, RFP and EMB so far have been known as 0.025-0.05, 0.005-0.5 and 0.5-8 ug/ml, respectively, and the MIC of PZA related cellular power of hydrogen is 1.5 ug/ml in the condition of pH 5.0 (1). The average concentrations of INH, RPF, PZA and EMB in vertebral focus tissue were all lower than the corresponding MBC, which means the effect of conservative treatment was not good for these patients with sclerotic bone in the vertebral focus; moreover, they were prone to drug-resistant tuberculosis."

Minor comment 1: You state the MICs of the drugs in this paragraph but then refer to MBC of the drugs to compare to the concentration of drugs found in sclerotic tissue -- is there a difference between MIC and MBC used in the paper? If so, explicitly state how they are different. If not, pick one and consistently use it throughout the paper. 

Minor comment 2: The sentence 'cellular power of hydrogen' makes no sense to me --please rephrase: "MIC of PZA related cellular power of hydrogen is 1.5 ug/ml in the condition of pH 5.0"

Author Response

Reviewer 1s Comments

1.This is a vastly improved version of the previous version. I commend the authors for making the recommended changes, which as improved the clarify of the manuscript.

Responds: Thanks for your comment. We are very cheerful for your approval to our revision and thank you for your valuable efforts and professional review to improve the quality of our manuscript.

2.In the below sentence you compares MICs of the drugs to concentrations of the same drugs found in sclerotic tissue. My concern is that MIC are presented in units of ug/ml whereas drug concentrations in sclerotic tissue are presented in different units (ug/mg). Since bone tissue consists of mostly dead weight in my opinion (calcium-phosphate) the effective concentration of the drug is likely much higher. Therefore I'm not sure of the soundness of making a direct comparison between known MICs and concentration of drug found in sclerotic tissue to conclude that the drugs are ineffective. Further you write that "The effect of conservative treatment was not good for these patients with sclerotic bone in the vertebral focus; moreover, they were prone to drug-resistant tuberculosis” -- but in the materials and method you write, "All patients have been fully recovery with no STB recurrence after one year fellow-up.” These two sentences are incompatible. However, the fact that all patients fully recovered does support the notion that the effective concentration of anti-TB drugs in sclerotic tissue was sufficient for curative treatment, which goes against your hypothesis. Please explain. Finally, for the sentence "they were prone to drug-resistant tuberculosis" where is the data in the paper to support this? 

Responds: Thanks for your comments. There are my response:

(a) The unit of MIC and sclerotic tissueare different as ug/ml and ug/mg  In our sample processing procedure, we turned the bone tissue into bone homogenate (dilution ratio is 1: 10) with the unit of µg/ml and then apply it to the HPLC/MSMS to get the drugs concentration whose unit is µg/mg. In this transform procedure, by the calculation we can get that 1 µg/mg (bone tissue) = 1 µg/ml (plasma, MIC), therefore the concentration is theoretically comparative between sclerotic tissue and MIC which Ge et al. and Liu et al. have already compared [1, 2]. We agree with your opinion that due to that bone tissue consists of mostly dead weight, the drugs concentration is underestimate despite that we use the liquid nitrogen to ryodesiccate bone tissue and pulverize it as efficient as we can to undermine the measuring error in the sample processing. Thanks for your reminder, we have realized that the conclusion of direct comparison between MICs and drug concentration in sclerotic tissue is limited convincing, therefore we have revised the sentence as “The average concentrations of INH, RPF, PZA and EMB in vertebral focus tissue were all lower than the corresponding MBC, which suggests that the effect of conservative treatment may be not good for these patients with sclerotic bone in the vertebral focus. However due to the bone tissue sample processing limitation the drugs concentration of vertebral focus tissue may be underestimate which need to be further validated in the future work. 

(b) We are sorry about the confusion of all STB patients are full recovery caused to you. All the included STB patients have undertook the focus debridement surgery, and there are several patients who is insensitive to the anti-tuberculosis chemotherapyso as to undertake surgery treatment, most of them belong to the sclerotic bone group. After the surgery these patients continued the previous anti-tuberculosis chemotherapy and got full recovery after one year fellow-up, the key step of the surgery is to remove the focus tissue of the vertebra including the sclerotic bone. The full recovery of the STB patients with sclerotic bone who was insensitive to the anti-tuberculosis chemotherapy before can welly prove the fact that the removal of sclerotic bone can improve the curative effect of anti-tuberculosis chemotherapy which is consist with my conclusion. Therefore the effect of conservative treatment was not good for these patients with sclerotic bone because the sclerotic bone can hinder the drugs concentration, once the sclerotic bone is removed the curative effect of anti-tuberculosis chemotherapy can be improved for full recovery.

(c) Thanks for the kind reminder, we found that the conclusion of “they were prone to drug-resistant tuberculosis” is poor supported in the manuscript, therefore we have deleted it from the manuscript for no more misleading.

Reference

[1] Ge, Z.;Wang, Z.; Wei,  Measurement of the concentration of three antituberculosis drugs in the focus of spinal tuberculosis. Eur Spine J. 2008, 17, 1482-7.

[2] Liu,P.; Zhu, Q.; Jiang,  Distribution of three antituberculous drugs and their metabolites in different parts of pathological vertebrae with spinal tuberculosis. Spine. 2011, 36, E1290-5.

3.You state the MICs of the drugs in this paragraph but then refer to MBC of the drugs to compare to the concentration of drugs found in sclerotic tissue -- is there a difference between MIC and MBC used in the paper? If so, explicitly state how they are different. If not, pick one and consistently use it throughout the paper.

Responds: Thanks for your comments. We picked the MIC consistently throughout the paper and deleted the MBC in the revised manuscript. It is widely accepted that an effective bactericidal concentration level should be 10 times of the MIC level. In order to better clarify the concentration relationship, we revised the sentence as It is widely accepted that an effective bactericidal concentration level should be 10 times of the MIC level. The average concentrations of INH, RPF, PZA and EMB in vertebral focus tissue were all lower than the corresponding effective bactericidal concentration and most lower than the corresponding MIC, which suggests that the effect of conservative treatment may be not good for these patients with sclerotic bone in the vertebral focus. 

Reference

[1] Xie HA, Yang GT, Lin SZ, et al. Contemporary Phthisiology. Beijing: People’s  Health Publishing House; 2000: 510–27.

4.The sentence 'cellular power of hydrogen' makes no sense to me --please rephrase: "MIC of PZA related cellular power of hydrogen is 1.5 ug/ml in the condition of pH 5.0"

Responds: Thanks for your comments. We have revised the sentence as “MIC of PZA is 1.5 ug/ml in the condition of pH 5.0”

Reviewer 3 Report

Thank you for addressing my comments.  More information on the LC/MS/MS assay would be beneficial to readers, suggest adding MRM transitions.

I support publishing this version.

Well done with excellent work and a great contribution to the field.

Author Response

Reviewer 3s Comments

1.Thank you for addressing my comments. More information on the LC/MS/MS assay would be beneficial to readers, suggest adding MRM transitions..

Responds: Thanks for your comment. The MRM transitions of four drugs in LC/MS/MS are as follows:

EMB: m/z 205.2116.1

INH: m/z 138.1121.1

PZA: m/z 124.181.1

RIF: m/z 823.4791.4

We have added the MRM transitions detail in the methods part in the manuscript.

2.I support publishing this version. Well done with excellent work and a great contribution to the field.

Responds: Thanks for your comment. We are very cheerful for your approval to our revision and thank you for your valuable efforts and professional review to improve the quality of our manuscript.
